Effect of foliar application of microelements on chlorophyll content, canopy architecture indicators, and physiological parameters of Hordeum vulgare L. plants

Stadnik Barbara 1 2 barbarast@dokt.ur.edu.pl
Tobiasz-Salach Renata 2
Migut Dagmara 3
1 Doctoral School of the University of Rzeszów, University of Rzeszów , Rzeszów , Poland
2 Department of Crop Production, University of Rzeszów , Rzeszów , Poland
3 Faculty of Biotechnology, Collegium Medicum, University of Rzeszów , Rzeszów , Poland
Brygadyrenko Viktor
Electronic publication date: 2025 Sep 18
Publication date: 2025
Volume: 13
Electronic Location ID: e19966
Received 2025 Apr 18; Accepted 2025 Jul 31
Copyright: © 2025 Stadnik et al.
Copyright year: 2025
Copyright holder: Stadnik et al.
License: This is an open access article distributed under the terms of the Creative Commons Attribution License, which permits unrestricted use, distribution, reproduction and adaptation in any medium and for any purpose provided that it is properly attributed. For attribution, the original author(s), title, publication source (PeerJ) and either DOI or URL of the article must be cited.
License URL: https://creativecommons.org/licenses/by/4.0/

Keywords: Barley, Chlorophyll fluorescence, Foliar fertilization, Gas exchange, Microelement fertilizer

Funding: Ministry of Science and Higher Education Department of Crop Production, Faculty of Technology and Life Sciences, University of Rzeszów This research was financed from the funds of the Ministry of Science and Higher Education for the scientific activities of the Department of Crop Production, Faculty of Technology and Life Sciences, University of Rzeszów. The funders had no role in study design, data collection and analysis, decision to publish, or preparation of the manuscript.

==============================
Background

The availability of nutrients determines the proper development and yield of plants. Microelements play an important role primarily as cofactors of important enzymes in the metabolic and physiological processes of plants. Barley (Hordeum vulgare L.) is one of the most popular cereals in the world. It is used mainly as animal feed and as a raw material for the brewing industry.

Methods

A three-year field experiment was conducted to determine the effect of foliar spraying of plants with single-component fertilizers containing copper, manganese, molybdenum, and zinc on the physiological parameters of barley plants. In the experiment, three malting barley cultivars were tested: Baryłka, KWS Irina, RGT Planet. During the vegetation period of the plants after foliar fertilization, canopy architecture indices, chlorophyll content in leaves, and gas exchange and chlorophyll fluorescence parameters were measured.

Results

The foliar application of each of the microelements tested caused a significant increase in the chlorophyll content in the leaves and canopy architecture parameters (LAI and MTA) compared to the control. Furthermore, a significant increase in gas exchange parameters was also observed: intercellular CO2 concentration (Ci), transpiration rate (E), stomatal conductance (gs), net photosynthetic rate (PN) and chlorophyll fluorescence parameters, maximum quantum yield of PSII photochemistry (Fv/Fm), fraction of active PSII reaction centers (RC/ABS), potential photochemical efficiency (Fv/F0), and performance index (PI). The highest results of physiological parameters such as CCI, Ci, E, Fv/Fm, Fv/F0 and PI were recorded after the application of zinc fertilizer. LAI and MTA were highest in plots with plants sprayed with molybdenum-based fertilizer. The foliar application of Cu caused the greatest increase in Ci and PN. The experiment showed significant genotypic differences. The RGT Planet cultivar had a higher chlorophyll content, E and PN parameters among the tested cultivars. The KWS Irina cultivar was characterized by the highest values of canopy architecture parameters and significantly lower values of CCI, Fv/Fm and Fv/F0 compared to the Baryłka and RGT Planet.

Introduction

The use of micronutrients is an important aspect in plant nutrition. All higher plants require six essential micronutrients such as manganese, iron, copper, zinc, boron, and molybdenum. These elements are essential components, required in small amounts for proper plant growth and development (Welch & Shuman, 1995; Ram et al., 2017; Kaur et al., 2023). Microelements are involved in various metabolic processes. The use of micronutrients increases antioxidant metabolism in plants and helps alleviate environmental stresses (Ahmed et al., 2020; Tavanti et al., 2021; Vatansever, Ozyigit & Filiz, 2017). However, too high doses of micronutrients can be toxic to plants (Gupta, 2010; Cruz et al., 2022). Optimizing plant nutrition with mineral compounds is a sustainable approach to improve plant health and yields. The supply of nutrients through foliar fertilization is a faster and more environmentally friendly method compared to soil fertilization. Nutrients can be applied in controlled amounts and at a specific time of plant growth (Niu et al., 2021). Furthermore, foliar fertilization is particularly effective in situations where nutrient uptake by the root system is limited, for example, in conditions of drought, excessive soil compaction, or oxygen deficiency in the rhizosphere. Currently, the foliar fertilization strategy is used in agriculture as a form of complementary plant nutrition, especially in crops with a high potential for yield (Niu et al., 2021; Oliveira et al., 2022; Januszkiewicz, Kulczycki & Samoraj, 2023). Barley (Hordeum vulgare L.) ranks fourth among all cereal crops and is widely grown for human consumption, as feed for livestock and for brewing and malting purposes around the world (Meints & Hayes, 2019; FAO, 2025).

Barley requires micronutrients such as copper (Cu), manganese (Mn), molybdenum (Mo), and zinc (Zn) for optimal growth and development, although the significance of specific elements may vary depending on soil properties and environmental conditions (Afzal et al., 2020; Assunção et al., 2022; Nyiraguhirwa et al., 2022). In some agroecosystems, boron (B) or iron (Fe) may also be critical. Effective micronutrient management in barley involves a combination of soil and foliar applications, balanced with macronutrient fertilization. Regular soil testing and tailored fertilization strategies are essential to address specific nutrient deficiencies and optimize barley production (Mazur et al., 2024; Grzanka et al., 2024).

Micronutrients such as copper (Cu) and manganese (Mn) play key roles in plant metabolism, participating in electron transport chains during photosynthesis and respiration and acting as cofactors for numerous antioxidant and defense enzymes (Yruela, 2009; Mir, Pichtel & Hayat, 2021). Manganese plays a crucial role in the functioning of photosystem II, where it catalyzes water splitting and supplies electrons to the photosystem. Although the importance of these elements is well documented, much less is known about their effects under foliar application conditions in the field, particularly with respect to photosynthetic parameters and gas exchange in cereals. Manganese deficiency negatively affects plants by limiting the functioning of key enzymes responsible for the removal of reactive oxygen species, seed germination, and photosynthesis (Waldron et al., 2009; Stoltz & Wallenhammar, 2014; Schmidt & Husted, 2019).

Also, selective enzymes use molybdenum to carry out redox reactions. Plant molybdenum enzymes are involved in nitrogen reduction and assimilation, and sulfate metabolism. Molybdenum fertilization by foliar sprays can effectively replenish internal molybdenum deficiencies and support molybdenum enzyme activity (Kaiser et al., 2005; Rana et al., 2020). Zinc is a catalytic and structural protein cofactor in hundreds of enzymes and performs key structural functions in protein domains that interact with other molecules. The largest class of Zn-binding proteins in organisms is the zinc finger domain (Broadley et al., 2007; Rudani, Vishal & Kalavati, 2018; Cabot et al., 2019; Stanton et al., 2022). Photosynthesis is one of the most important biochemical pathways by which plants convert solar energy into chemical energy and grow. It is the most basic and complex physiological process in all green plants (Evans, 2013; Janssen et al., 2014; Ashraf & Harris, 2013). It is the basic physiological activity that determines the formation of plant yield, and the strength of photosynthesis is closely related to the level of yield (Lawlor, 1995; Long et al., 2006; Zhang et al., 2022). Analysis of chlorophyll fluorescence parameters is a useful technique that reflects the condition of plants. It plays an important role in understanding the basic mechanisms of photosynthesis and the response of plants to environmental changes. Chlorophyll content and chlorophyll fluorescence are important markers of the photosynthetic state of the plant (Maxwell & Johnson, 2000; Murchie & Lawson, 2013; Goltsev et al., 2016; Kalaji et al., 2016, 2018). Previous studies have rarely comprehensively assessed the effects of foliar micronutrient applications on barley physiology in intensive cropping systems typical of temperate climate zones. In particular, data on the effects of these treatments on chlorophyll fluorescence and gas exchange parameters are lacking, limiting the ability to fully understand their mechanisms of action in agricultural practice. Therefore, there is growing interest in assessing the physiological responses of cereal plants to foliar micronutrients, particularly in the context of photosynthetic efficiency and adaptation strategies in temperate climates. The foliar application of micronutrients has the potential to precisely and rapidly replenish element deficiencies in critical plant development stages (Fernández & Brown, 2013). However, the application of this technique in malting barley cultivation requires a thorough assessment of plant physiological responses, such as photosynthesis rate, chlorophyll content, transpiration rate, and chlorophyll fluorescence parameters. These parameters are rarely studied in field studies. Previous publications indicate that adequate micronutrient supply can significantly influence photosynthesis rate, water management efficiency, enzyme activity, and plant resistance to abiotic stresses (Niu et al., 2021; Kaya & Ashraf, 2024). However, comprehensive studies linking these aspects to foliar fertilization in malting barley cultivation are lacking. The study was carried out to investigate the effect of the foliar application of selected microelements on the chlorophyll content, canopy architecture, and physiological parameters of barley plants. The research hypothesis assumes that the application of microelements will have a positive effect on the course of the analyzed photosynthesis parameters of barley plants.

Materials and Methods

A three-year field experiment (2019–2021) was conducted in Reczpol (49°47′03″N, 22°34′37″E), in southeastern Poland, according to the methodology described by Stadnik, Tobiasz-Salach & Migut (2024). The experiment was set up in a split-block design with four replications. The main factor (block) was a spring barley cultivar (Baryłka, KWS Irina, RGT Planet), while the effect of a subfactor, i.e., variants of foliar fertilization with micronutrients (Cu, Mn, Mo, Zn and the non-fertilization control), was analyzed within each cultivar. The area of a single crop plot was 15 m2. Agrotechnical treatments were conducted according to intensive spring barley cultivation technology, including full protection of the plant against weeds, diseases, and pests. Winter oil seed rape was the preceding crop.

In each study year, certified source seed material was seeded in the first ten days of April (6 April 2019, 2 April 2020, 8 April 2021), at a rate of 165 kg ha−1, in rows 12 cm apart, at a depth of 3 cm. Harvest was carried out at full grain maturity (BBCH 89) (Lancashire et al., 1991) in the third decade of July (July 22, 2019, July 30, 2020, July 25, 2021).

Before the experiment, the physicochemical properties of the soil were analyzed in an accredited laboratory. The soil pH was determined potentiometrically (PN-ISO 10390:1997, 1997), organic carbon content using the Tiurin method (PN-ISO-14235:2003, 2003), available phosphorus and potassium using the Egner-Riehm method (PN-R-04023:1996, 1996; PN-R-04022:1996+AZ1:2002, 1996) and magnesium using the Schachtschabel method (PN-R-04020:1994+AZ1:2004 (1994)). The trace elements (Cu, Mn, Zn, and Fe) were determined after mineralization in HCl using flame absorption spectrometry (PN-R-04017:1992, 1992; PN-R-04019:1993, 1993; PN-R-04016:1992, 1992; PN-R-04021:1994, 1994). The soil was slightly acidic, containing approximately 1% organic carbon. Cu content was low in the first year and average in subsequent years, while other trace elements were present at average levels. Mineral (pre-sowing) fertilization included 50 kg ha−1 N, 60 kg ha−1 P2O5, and 90 kg ha−1 K2O. Polifoska® 6 was applied at a rate of 300 kg ha−1 and ammonium nitrate (32% N) at 100 kg ha−1. During the tillering phase (BBCH 25–27), a multi-component foliar fertilizer, Opti Zboża (3 kg ha−1), containing macro- and micronutrients, was additionally applied.

The experimental factors were the following: I. Barley varieties: Baryłka (Hodowla Roślin Strzelce Sp. z o.o., IHAR Group), KWS Irina (KWS Lochow GmbH), RGT Planet (RAGT 2n);

II. Foliar fertilization with micronutrients: Control (no fertilization), Cu (MIKROVIT® COPPER), Mn (MIKROVIT® MANGANESE), Mo (MIKROVIT® MOLYBDENUM), Zn (MIKROVIT® CYNK) (Table 1).

Table 1 Characteristics of foliar fertilizers applied in the field experiment with spring barley (2019–2021).

Trade name	Active ingredient	Chemical form	Application rate	BBCH stage	Method of application	
MIKROVIT® COPPER	75 g Cu·L−1	Copper sulfate (CuSO4)	2 L·ha−1	BBCH 30–32	Aqueous foliar spraying	
MIKROVIT® MANGANESE	160 g Mn·L−1	Manganese sulfate (65 g) + manganese nitrate (95 g)	2 L·ha−1	BBCH 30–32	Aqueous foliar spraying	
MIKROVIT® MOLYBDENUM	33 g Mo·L−1	Ammonium molybdate ((NH4)6Mo7O24)	1 L·ha−1	BBCH 30–32	Aqueous foliar spraying	
MIKROVIT® ZINC	112 g Zn·L−1	Zinc sulfate (ZnSO4)	2 L·ha−1	BBCH 30–32	Aqueous foliar spraying	

The application was performed once during the stem elongation phase (BBCH 30–32) (Lancashire et al., 1991) in windless conditions, using a pressure sprayer. Grain yield was assessed separately for each plot.

The mean temperatures and total rainfall during the growing season were determined based on data from the meteorological station in Zadąbrowo (50°19′09″N, 21°48′19″E). The Sielianinow hydrothermal coefficient (k) was calculated to assess water-thermal conditions. Based on the k value, meteorological conditions were characterized as dry, optimal, humid, or extreme (Stadnik, Tobiasz-Salach & Migut, 2024).

Measurements of chlorophyll content, canopy architecture indices, and plant physiological parameters

In each year, measurements were taken twice during the plant vegetation period in BBCH 43-45 and BBCH 50-52 phases (Lancashire et al., 1991). The results are presented as averages of two measurement dates.

Chlorophyll content index measurement

The measurement of chlorophyll content index (CCl) content was performed using a CCM-200plus handheld chlorophyll meter (Opti-Sciences, Hudson, NH, USA). Five measurements were made on the subflag leaf of random plants in each plot (Stadnik, Tobiasz-Salach & Mazurek, 2022).

Measurement of canopy architecture indicators

Leaf area index (LAI) and mean tilt angle (MTA) measurements were recorded with an LAI-2000 instrument (LI-COR, Lincoln, NE, USA). The LAI measurement was made in each plot by performing one measurement on the canopy and four in the canopy (Jańczak-Pieniążek & Kaszuba, 2024).

Measurement of gas exchange parameters

The photosynthesis measurement system LCpro-SD (ADC Bioscientific Ltd., Herts, UK) was used to measure the photosynthesis of leaves. The following parameters were tested: intercellular CO2 concentration (Ci), transpiration rate (E), stomatal conductance (gs), and net photosynthetic rate (PN). The LCpro-SD plant leaf photosynthesis chamber has a flow accuracy of ±2% of its range. During measurement, the light intensity in the measuring chamber was 350 µmol m−2 s−1, and the temperature was approximately 22 °C, with a relative humidity of 60%. Measurements were taken on three randomly selected subflag leaves in each plot (Stadnik, Tobiasz-Salach & Mazurek, 2022).

Chlorophyll fluorescence parameter measurement

Chlorophyll fluorescence was measured using a Pocket PEA continuous excitation fluorimeter (Pocket PEA; Hansatech Instruments, King’s Lynn, Norfolk, UK) equipped with black shade clamps that were applied to the leaf blade away from the leaf vein. The leaves were adapted to the dark for a period of 30 min. The fluorescence signal was collected in red actinic light with a maximum light source wavelength of 627 nm and transmitted for 1 s at a maximum available intensity of 3,500 μmol (photon) for photosynthetically active radiation (PAR) m−2 s−1. Measurements were taken on five randomly selected subflag leaves in each plot. The following parameters were measured: maximal quantum yield of PSII photochemistry (Fv/Fm), maximum quantum yield of primary photochemistry (Fv/F0), fraction of active PSII reaction centers (RC/ABS), and photosynthetic efficiency index (PI) (Stadnik, Tobiasz-Salach & Mazurek, 2022).

Statistical analysis and correlation coefficient

Data from the three-year study were subjected to split-block analysis of variance (ANOVA) using TIBCO Statistica 13.3.0 statistical software. A synthesis of results was performed using a cross-classified hierarchical model, with the year treated as a fixed factor. The structure of the statistical model was aligned with the organization of individual field trials, which enabled the standardization of the comparison schemes and consideration of the seasonal variation in the meteorological conditions, as well as interactions among the experimental factors. Before the analysis was performed, the assumptions of ANOVA were verified. These included the normality of residuals, tested using the Shapiro–Wilk test, and the homogeneity of variance, verified using the Levene test. The mean values were calculated and the significance of the differences between groups was assessed using Tukey’s honestly significant difference test at a significance level of p ≤ 0.05. All quantitative variables analyzed, including yield and twelve physiological parameters, were complete and included 180 observations. Measurements were conducted in 45 unique combinations of the factors of year, cultivar, and micronutrient treatment, which determined the number of comparative units used in the ANOVA.

The relationships between yield and physiological parameters were evaluated using Pearson correlation coefficients. Data were aggregated at the mean level for each combination of cultivar, treatment, and year, preserving variability related to genotype and growing season while minimizing the risk of overestimating the effective sample size. As a result, the final number of observations used for the correlation analysis was 45. All quantitative variables were previously verified for compliance with a normal distribution.

Results

Chlorophyll content index, LAI, and MTA

The experiment carried out demonstrated a significant effect of the application of microelements in the foliar environment on the chlorophyll content and the canopy architecture parameters in the spring barley cultivars tested (Table 2). The average CCI value in the experiment was 40.6. The highest CCI values were recorded in the Baryłka and KWS Irina cultivars after the application of Cu (by 14.4% and 6.4%, respectively, compared to the control), while in the RGT Planet cultivar the height effect was achieved by the application of Zn in the foliar (an increase of 13.0% compared to the control) (Table 2). Irrespective of the foliar fertilization used, statistical analysis showed a higher CCI value in the Baryłka and RGT Planet cultivars compared to the KWS Irina cultivar. The application of each of the microelement fertilizers tested (irrespective of the cultivar) influenced the increase in CCI compared to the control. The use of Cu resulted in an increase in CCI in barley leaves on an average of 8.9%, Mn–5.2%, Mo–8.1%, and Zn–8.6% (Table 2). The study assessed the values of the canopy architecture indices based on experimental factors. The highest LAI values were obtained in the Baryłka and KWS Irina cultivars after Mo application (an increase of 11.7% and 10.8%, respectively, compared to the control), but the increase was not statistically confirmed. In the RGT Planet cultivar, in the plots where Mn and Mo foliar fertilization was applied, a significant increase in LAI was observed compared to the control, while the application of Zn caused a decrease in the measured parameter.

Table 2 Effect of foliar fertilization and cultivar on the chlorophyll content in the leaves and the parameters of the barley canopy architecture.

Cultivar (C)	Fertilization (F)	CCI	LAI (m2 m−2)	MTA (°)	
Baryłka	Control	36.9 ± 8.2a	4.69 ± 0.38a	42.5 ± 2.6a	
Cu	42.2 ± 9.4a	5.22 ± 0.55a	46.4 ± 4.5a	
Mn	40.9 ± 8.3a	4.88 ± 0.34a	43.8 ± 3.6a	
Mo	41.2 ± 8.3a	5.24 ± 0.47a	45.8 ± 5.6a	
Zn	40.6 ± 7.8a	5.14 ± 0.40a	42.6 ± 3.1a	
KWS Irina	Control	39.2 ± 5.5a	4.91 ± 0.62a	41.4 ± 2.4a	
Cu	41.7 ± 5.9a	5.14 ± 1.11a	45.3 ± 4.5a	
Mn	38.6 ± 6.2a	5.08 ± 0.85a	47.8 ± 5.6a	
Mo	39.7 ± 7.3a	5.44 ± 0.83a	53.0 ± 8.3b	
Zn	40.5 ± 8.7a	5.38 ± 1.14a	50.5 ± 7.6ab	
RGT Planet	Control	38.5 ± 7.6a	5.07 ± 0.97a	46.6 ± 7.9a	
Cu	41.0 ± 8.7a	5.04 ± 0.52a	48.4 ± 6.2a	
Mn	41.2 ± 9.4a	5.12 ± 0.81a	47.4 ± 4.5a	
Mo	43.1 ± 6.7b	5.12 ± 0.77a	45.5 ± 3.9a	
Zn	43.5 ± 11.6b	4.75 ± 0.58a	47.5 ± 4.6a	
HSDp ≤ 0.05 C×F	** F = 3.7, df = 8.354, p = 0.0026	n.s. F = 1.6, df = 8.354, p = 0.1310	*** F = 6.0, df = 8.354, p = 0.0000	
Mean	Baryłka	40.4 ± 8.6AB	5.04 ± 0.48AB	44.2 ± 4.3A	
KWS Irina	39.9 ± 6.9A	5.19 ± 0.93B	47.6 ± 7.2B	
RGT Planet	41.5 ± 9.1B	5.02 ± 0.75A	47.1 ± 5.6AB	
HSDp ≤ 0.05 C	* F = 3.9, df = 2.354, p = 0.0198	* F = 4.0, df = 2.354, p = 0.0186	** F = 6.4, df = 2.354, p = 0.0018	
Mean	Control	38.2 ± 7.2A	4.89 ± 0.71A	44.5 ± 5.4A	
Cu	41.6 ± 8.2B	5.13 ± 0.77AB	48.1 ± 5.2B	
Mn	40.2 ± 8.1B	5.03 ± 0.70AB	47.6 ± 4.9B	
Mo	41.3 ± 7.6B	5.26 ± 0.71C	49.5 ± 7.0C	
Zn	41.5 ± 9.6B	5.09 ± 0.81AB	47.9 ± 6.33BC	
HSDp ≤ 0.05 F	*** F = 5.9, df = 4.354, p = 0.0000	*** F = 4.5, df = 4.354, p = 0.0004	*** F = 5.9, df = 4.354, p = 0.0000	
Year (Y)	2019	41.9 ± 7.8B	4.87 ± 0.41B	49.3 ± 5.4C	
2020	33.3 ± 5.7A	4.65 ± 0.53A	47.4 ± 6.6B	
2021	46.6 ± 4.6C	5.72 ± 0.76C	42.1 ± 2.9A	
HSDp ≤ 0.05 Y	*** F = 110.1, df = 2.1170, p = 0.0000	*** F = 215.7, df = 2.1170, p = 0.0000	*** F = 187.2, df = 2.1170, p = 0.0000	
Mean	40.6 ± 8.3	5.08 ± 0.75	46.3 ± 6.0	
Note:

Asterisks (*, **, ***) indicate significant differences at 5% (p ≤ 0.05), 1% (p ≤ 0.01), and 0.1% (p ≤ 0.001) levels, respectively; n.s. – not significant, according to Tukey’s honestly significant difference test (HSD). Mean values followed by different letters (a–b and A–C) within a column differ significantly at p ≤ 0.05. Statistical analysis based on a split-block ANOVA model.

Regardless of the application of the microelement, the KWS Irina cultivar was shown to be characterized by a higher LAI value compared to the other cultivars tested (Baryłka and RGT Planet).

The application of each of the microelement fertilizers tested (irrespective of the cultivar) caused a significant increase in LAI compared to the control. The highest average value was recorded after Mo application (an increase of 7.6%) (Table 2).

The value of the MTA index was the highest in the KWS Irina cultivar after Mo application and was 53°. In the Baryłka and RGT Planet cultivars, the highest MTA values were recorded in Cu plant fertilization. The KWS Irina cultivar had 3.4° higher MTA, and the RGT Planet cultivar 2.9° higher MTA compared to the Baryłka (an increase of 7.7% and 6.6%, respectively). The highest average MTA value (irrespective of the cultivar) was recorded after Mo application (5° higher compared to the control −11.2%) (Table 2). Analysis of the results of subsequent years of the study showed a significant effect of the year of vegetation on the chlorophyll content in the leaves and the parameters of the architecture of the barley canopy. In 2021, CCI and LAI were the highest, while MTA was the lowest compared to previous years in which the field experiment was carried out (Table 2).

Gas exchange

In the experiment carried out, the values of the gas exchange parameters depended on the experimental factors applied (Table 3). In each of the cultivars tested, the Ci values were the highest in the plots fertilized with Cu or Zn. In the Baryłka cultivar, in these fertilization variants, a 3.5% increase in Ci was observed compared to the control, while in the KWS Irina cultivar, an increase of 3.8%. The RGT Planet cultivar, after the application of Cu and Zn, had a Ci value higher by 5.9% and 4.2%, respectively, compared to the control. Statistical analysis showed that fertilization (irrespective of cultivar) with each of the microelements tested caused a significant increase in Ci compared to the control. The highest values were observed with the application of Cu and Zn (an increase of 4.2% and 3.8%, respectively) (Table 3).

Table 3 Effect of foliar fertilization and cultivar on gas exchange parameters.

Cultivar (C)	Fertilization (F)	Ci (µmol (CO2) mmol−1)	E (mmol (H2O) m−2 s−1)	gs (mmol(H2O)m−2s−1)	PN (µmol (CO2) m−2 s−1)	
Baryłka	Control	287 ± 9ab	4.12 ± 0.45a	0.52 ± 0.06a	12.1 ± 1.8a	
Cu	297 ± 10a–d	4.48 ± 0.48ab	0.53 ± 0.08a	14.1 ± 1.8a–e	
Mn	292 ± 15a–c	4.34 ± 0.88ab	0.53 ± 0.07a	13.4 ± 0.8a–c	
Mo	293 ± 14a–c	4.42 ± 0.52ab	0.54 ± 0.04a	13.5 ± 1.4a–d	
Zn	297 ± 8a–d	4.81 ± 0.59ab	0.56 ± 0.08ab	14.2 ± 1.6a–e	
KWS Irina	Control	287 ± 12ab	4.30 ± 0.72a	0.53 ± 0.05a	13.2 ± 1.3ab	
Cu	298 ± 20b–d	4.80 ± 0.60ab	0.55 ± 0.05a	14.9 ± 1.3b–e	
Mn	293 ± 13a–d	4.99 ± 0.70b	0.58 ± 0.04b	15.1 ± 0.8b–e	
Mo	290 ± 29a–c	4.94 ± 0.91ab	0.57 ± 0.04b	14.9 ± 1.3b–e	
Zn	298 ± 11b–d	4.93 ± 0.77ab	0.56 ± 0.06ab	14.4 ± 1.6b–e	
RGT Planet	Control	287 ± 15a	4.70 ± 0.54ab	0.55 ± 0.06a	14.6 ± 2.3b–e	
Cu	304 ± 16d	4.97 ± 0.56b	0.54 ± 0.07a	16.1 ± 2.3f	
Mn	296 ± 13a–d	5.01 ± 0.38b	0.53 ± 0.06a	15.4 ± 1.8c–e	
Mo	294 ± 11a–d	5.04 ± 0.35b	0.54 ± 0.05a	15.5 ± 1.3c–e	
Zn	299 ± 10cd	5.08 ± 0.52b	0.54 ± 0.05a	15.6 ± 1.1d–e	
HSDp ≤ 0.05 C × F	n.s. F = 0.9, df = 8.61, p = 0.4859	** F = 1.6, df = 8.104, p = 0.0024	*** F = 1.1, df = 8.104, p = 0.0000	** F = 0.5, df = 8.104, p = 0.0020	
Mean	Baryłka	293 ± 12A	4.43 ± 0.64A	0.53 ± 0.07A	13.5 ± 1.7A	
KWS Irina	293 ± 19A	4.79 ± 0.79B	0.56 ± 0.05B	14.5 ± 1.4B	
RGT Planet	296 ± 14A	4.96 ± 0.49C	0.54 ± 0.06A	15.4 ± 1.9C	
HSDp ≤ 0.05 C	n.s. F = 3.8, df = 2.61, p = 0.0857	*** F = 3.0, df = 2.104, p = 0.0001	*** F = 4.3, df = 2.104, p = 0.0004	*** F = 4.7, df = 2.104, p = 0.0000	
Mean	Control	287 ± 13A	4.37 ± 0.62A	0.53 ± 0.06A	13.3 ± 2.1A	
Cu	299 ± 16C	4.75 ± 0.59B	0.54 ± 0.07AB	15.0 ± 2.0C	
Mn	294 ± 14B	4.78 ± 0.75B	0.55 ± 0.07AB	14.6 ± 1.5B	
Mo	292 ± 20B	4.80 ± 0.69B	0.55 ± 0.05B	14.6 ± 1.6B	
Zn	298 ± 10C	4.94 ± 0.64C	0.55 ± 0.06B	14.7 ± 1.6B	
HSDp ≤ 0.05 F	*** F = 20.5, df = 4.61, p = 0.0000	*** F = 13.7, df = 4.104, p = 0.0000	** F = 10.4, df = 4.104, p = 0.0097	*** F = 3.1, df = 4.104, p = 0.0000	
Year (Y)	2019	294 ± 11B	4.56 ± 0.65B	0.53 ± 0.06A	14.9 ± 1.1B	
2020	283 ± 15A	4.32 ± 0.46A	0.53 ± 0.05A	12.9 ± 1.5A	
2021	306 ± 11C	5.31 ± 0.50C	0.58 ± 0.06B	15.7 ± 1.7C	
HSDp ≤ 0.05 Y	*** F = 305.7, df = 2.1077, p = 0.0000	*** F = 182.3, df = 2.1077, p = 0.0000	*** F = 279.6, df = 2.1077, p = 0.0000	*** F = 33.8, df = 2.1077, p = 0.0000	
Mean	294 ± 15	4.73 ± 0.69	0.54 ± 0.06	14.5 ± 1.9	
Note:

Asterisks (*, **, ***) indicate significant differences at 5% (p ≤ 0.05), 1% (p ≤ 0.01), and 0.1% (p ≤ 0.001) levels, respectively; n.s. – not significant, according to Tukey’s honestly significant difference test (HSD). Mean values followed by different letters (a–b and A–C) within a column differ significantly at p ≤ 0.05. Statistical analysis based on a split-block ANOVA model.

The value of the transpiration rate E was the highest in the RGT Planet cultivar after Zn application (an increase of 8.1% compared to the control). A significantly higher value compared to the control (16.0%) was also observed for the KWS Irina cultivar when fertilized with Mn (Table 3). The KWS Irina and RGT Planet cultivars were characterized by a higher average E value compared to the Baryłka cultivar (0 8.1% and 12.0%, respectively). The application of each of the microelements tested on the leaf (irrespective of the cultivar) caused a significant increase in E compared to the control—the highest increase (by 13.0%) was observed with the application of Zn. The average value of gs in the experiment was 0.54 (Table 3). The highest gs values were observed for the KWS Irina cultivar after Mn and Mo application (an increase of 9.4% and 7.5%, respectively, compared to the control). In Baryłka and RGT Planet, no significant differences were shown after the application of microelements compared to the control. The application of microelements to the foliage (irrespective of the cultivars tested) caused a significant increase in gs compared to the control after the application of Mo and Zn (an increase in both variants of 3.4%).

The experimental factors applied significantly modified the value of the PN parameter (Table 3). The highest value of the measured parameter was recorded in the RGT Planet cultivar after the application of Cu (an increase of 10.3% compared to the control). The analysis of this gas exchange parameter (irrespective of foliar fertilization) showed that its value in the RGT Planet cultivar was higher compared to Baryłka and KWS Irina by 14.1% and 6.2%, respectively (Table 3).

Foliar fertilization with each of the microelements tested (irrespective of the cultivar) had a positive effect on the value of PN and caused a significant increase compared to the control. The highest increase–12.8%—was recorded after spraying the plants with Cu.

In the studies carried out, significant differences were observed depending on the year of the field experiment. The highest values of the gas exchange parameters were observed in 2021, and the lowest in 2019 (Table 3). In June 2019, the lowest level of precipitation and the highest average daily temperature were recorded compared to 2020 and 2021 (Stadnik, Tobiasz-Salach & Migut, 2024). This resulted in drought, which could have a negative impact on the course of physiological processes in barley plants.

Chlorophyll fluorescence

The average value of the Fv/Fm was 0.82. The highest values were recorded in the KWS Irina cultivar after spraying the plants with Zn (0.84). Compared to the control, the increase was 5.0%. The Baryłka cultivar (irrespective of fertilization) obtained a significantly higher value of Fv/Fm by 1.2% compared to KWS Irina. The application of the foliage of each of the microelements tested (irrespective of the cultivar) caused an increase in Fv/Fm compared to the control of 3.8% (Table 4).

Table 4 Effect of foliar fertilization and cultivar on chlorophyll fluorescence parameters.

Cultivar (C)	Fertilization (F)	Fv/Fm	RC/ABS	Fv/F0	PI	
Baryłka	Control	0.80 ± 0.02ab	2.87 ± 0.48a	4.66 ± 0.44a	8.44 ± 1.31a	
Cu	0.83 ± 0.04ab	3.11 ± 0.57ab	4.68 ± 0.37a	10.38 ± 1.47ab	
Mn	0.83 ± 0.03ab	3.36 ± 0.54b	4.68 ± 0.41a	10.77 ± 1.29b	
Mo	0.84 ± 0.04ab	3.35 ± 0.56b	4.57 ± 0.55a	10.78 ± 1.26b	
Zn	0.84 ± 0.04b	3.26 ± 0.44ab	4.74 ± 0.34a	10.67 ± 1.21ab	
KWS Irina	Control	0.79 ± 0.04a	2.50 ± 0.43a	4.09 ± 0.85a	8.28 ± 0.86a	
Cu	0.82 ± 0.02ab	2.92 ± 0.57a	4.69 ± 0.43a	9.72 ± 1.08a	
Mn	0.82 ± 0.02ab	2.92 ± 0.45a	4.54 ± 0.48a	9.57 ± 1.14a	
Mo	0.82 ± 0.03ab	2.97 ± 0.36a	4.66 ± 0.49a	9.63 ± 1.25a	
Zn	0.83 ± 0.03ab	3.09 ± 0.55a	4.70 ± 0.40a	10.26 ± 1.64a	
RGT planet	Control	0.82 ± 0.02ab	2.95 ± 0.42a	4.62 ± 0.44a	9.14 ± 1.07a	
Cu	0.83 ± 0.02ab	2.71 ± 0.41a	4.67 ± 0.46a	9.35 ± 1.20a	
Mn	0.82 ± 0.02ab	2.85 ± 0.40a	4.69 ± 0.44a	9.46 ± 1.12a	
Mo	0.83 ± 0.03ab	3.18 ± 0.71ab	4.85 ± 0.46b	10.35 ± 2.06ab	
Zn	0.83 ± 0.02ab	2.92 ± 0.40a	4.75 ± 0.37ab	9.74 ± 1.51a	
HSDp ≤ 0.05 C × F	*** F = 2.2, df = 8.558, p = 0.0000	*** F = 2.6, df = 8.558, p = 0.0000	*** F = 2.1, df = 8.558, p = 0.0000	*** F = 1.8, df = 8.558, p = 0.0000	
Mean	Baryłka	0.83 ± 0.04B	3.19 ± 0.55B	4.67 ± 0.43B	10.21 ± 1.58B	
KWS Irina	0.82 ± 0.03A	2.88 ± 0.52A	4.54 ± 0.60A	9.49 ± 1.38A	
RGT planet	0.82 ± 0.02AB	2.92 ± 0.51A	4.72 ± 0.44B	9.61 ± 1.49A	
HSDp ≤ 0.05 C	** F = 3.0, df = 2.558, p = 0.0083	*** F = 4.1, df = 2.558, p = 0.0001	*** F = 6.0, df = 2.558, p = 0.0048	*** F = 14.2, df = 2.558, p = 0.0001	
Mean	Control	0.80 ± 0.03A	2.77 ± 0.49A	4.46 ± 0.66A	8.62 ± 1.15A	
Cu	0.83 ± 0.03B	2.91 ± 0.54B	4.68 ± 0.42B	9.82 ± 1.33B	
Mn	0.83 ± 0.03B	3.04 ± 0.52C	4.64 ± 0.45B	9.93 ± 1.32B	
Mo	0.83 ± 0.03BC	3.17 ± 0.58D	4.69 ± 0.51B	10.26 ± 1.63C	
Zn	0.83 ± 0.03C	3.09 ± 0.49CD	4.73 ± 0.37B	10.22 ± 1.51C	
HSDp ≤ 0.05 F	*** F = 8.4, df = 4.558, p = 0.0000	*** F = 11.3, df = 4.558, p = 0.0000	*** F = 4.6, df = 4.558, p = 0.0000	*** F = 22.7, df = 4.558, p = 0.0000	
Year (Y)	2019	0.84 ± 0.04C	2.90 ± 0.51B	4.90 ± 0.38C	9.46 ± 1.08B	
2020	0.81 ± 0.02A	2.83 ± 0.44A	4.40 ± 0.42A	8.99 ± 1.24A	
2021	0.82 ± 0.02B	3.26 ± 0.57C	4.63 ± 0.55B	10.86 ± 1.53C	
HSDp ≤ 0.05 Y	*** F = 133.1, df = 2.1117, p = 0.0000	*** F = 84.5, df = 2.1117, p = 0.0000	*** F = 141.7, df = 2.1117, p = 0.0000	*** F = 113.9, df = 2.1117, p = 0.0000	
Mean	0.82 ± 0.03	3.00 ± 0.54	4.64 ± 0.50	9.77 ± 1.52	
Note:

Asterisks (*, **, ***) indicate significant differences at 5% (p ≤ 0.05), 1% (p ≤ 0.01), and 0.1% (p ≤ 0.001) levels, respectively; n.s. – not significant, according to Tukey’s honestly significant difference test (HSD). Mean values followed by different letters (a–b and A–C) within a column differ significantly at p ≤ 0.05. Statistical analysis based on a split-block ANOVA model.

The RC/ABS parameter was also dependent on the experimental factors used. The highest values were recorded in the Baryłka cultivar with Mn and Mo fertilization (an increase of 17.1% and 16.7%, respectively, compared to the control). In the RGT Planet cultivar, an increase in this chlorophyll fluorescence parameter was observed after Mo application (an increase of 7.8% compared to the control), which was confirmed by variance analysis.

Regardless of fertilization, Baryłka was shown to have a higher average value of RC/ABS by 10.8% compared to KWS Irina and by 9.2% compared to RGT Planet. The foliar application of microelements in barley plants caused a significant increase in RC/ABS, and the highest average values were observed after spraying the plants with Mo and Zn (an increase of 14.4% and 11.6%, respectively, compared to the control) (Table 4).

The value of Fv/F0 was the highest in the RGT Planet cultivar after Mo and Zn application, while the lowest in KWS Irina in the control variant. The value of the parameter varied among the examined cultivars. The Baryłka and RGT Planet cultivars had significantly higher Fv/F0 values compared to the KWS Irina cultivar (by 2.9% and 4.0%, respectively). The foliar application of each of the microelement fertilizers tested caused a significant increase in Fv/F0 compared to the control. The best effect was observed after the application of Zn (an increase of 6.0%).

The highest PI values were observed in the Baryłka cultivar after the application of Mn and Mo. The increase compared to the control was 27.6% and 27.7%, respectively. The Baryłka cultivar (irrespective of foliar fertilization) was characterized by a significantly higher PI compared to the KWS Irina and RGT Planet cultivars (by 7.6% and 6.2%, respectively). The application of each of the microelement fertilizers caused a significant increase in PI compared to the control (irrespective of the cultivar). The greatest effect was observed after the application of Mo and Zn (with 16.0% and 15.7%, respectively) (Table 4).

The values of the chlorophyll fluorescence parameters varied depending on the year of the field experiment. The highest values of Fv/Fm and Fv/F0 were recorded in 2019, while RC/ABS and PI in 2021. In 2019, all measured chlorophyll fluorescence parameters were the lowest (Table 4). This year, at the time of the measurements, relatively high temperatures and low rainfall were recorded, which could have influenced the course of physiological processes in plants (Stadnik, Tobiasz-Salach & Migut, 2024).

Pearson’s correlation coefficient

The barley grain yield of the field experiment was presented in the publication by Stadnik, Tobiasz-Salach & Migut (2024). All correlation coefficients refer to aggregated data for application, operation, and year, with a sample size of n = 45. The table also presents the results of the correlations extracted from independent users and Fisher’s tests, which are distributed between the data sets (p ≤ 0.05 between all users). The strongest positive correlations with yield were recorded for transpiration rate (r = 0.81), chlorophyll content index (0.69), intercellular CO2 concentration (0.62), and net photosynthesis rate (0.57). This relationship between MTA and yield (r = −0.28) was not statistically significant (Fig. 1).

Figure 1 Correlation coercions (r) between grain yield and canopy architecture indices and plant physiological parameters.

The table presents Pearson correlation coefficients (r) between yield and physiological parameters; Asterisks (*, **, and ***) indicate significance at p ≤ 0.05, p ≤ 0.01, and p ≤ 0.001, respectively; n.s. denotes non-significant correlations (p > 0.05); all correlations were calculated based on n = 45, using mean values for each combination of year, cultivar, and micronutrient treatment; prior to analysis, variables were tested for compliance with the normal distribution.

Discussion

Plant growth and development depend on environmental conditions, including the availability of macro and microelements. Differences in nutrient content significantly affect the photochemical process of photosynthesis, therefore, they play a key role in biomass production (Kalaji et al., 2018; Cakmak & Engels, 2024). Micronutrients, although needed by plants in small amounts, play an important role in metabolic processes related to photosynthesis, chlorophyll formation, cell wall development, and respiration (Ram et al., 2017; Therby-Vale et al., 2022). Scientific studies on the effect of the foliar application of important micronutrients for plants focus primarily on their effect on mitigating environmental stresses and improving yield and its components (El-Magid, 2001; Ahanger et al., 2016; Kaur et al., 2023; Singhal et al., 2023). The aim of our study was to determine the effect of foliar fertilization with selected microelements on chlorophyll content, canopy architecture parameters, and the course of photosynthesis in spring barley under field conditions.

Leaf chlorophyll is a key indicator of leaf greenness, often used to determine nutrient deficiencies in leaves (Schlichting et al., 2015; Shah, Houborg & McCabe, 2017; Liu et al., 2019). In the field experiment conducted, the application of each of the microelements tested increased CCI in barley leaves, and the highest mean values were recorded in the plots fertilized with Cu and Zn (Table 1). Copper is an essential transition metal with redox activity, and it is involved in many physiological processes in plants. Cu deficiency affects young leaves, causing chlorosis and reduced photosynthetic activity. The positive effect of the foliar application of copper can be attributed to an important metabolic function, since this metal participates in photosynthesis and chloroplast development (Amberger, 1974; Yruela, 2005; El-Metwally et al., 2010; Yruela, 2013; Harangozo et al., 2017). Also, the study by Tobiasz-Salach & Augustyńska-Prejsnar (2020) conducted on barley, copper fertilization had a positive effect on chlorophyll content and, similarly to our study, the effect of copper was more visible than that of manganese. Improvements in physiological functions and an increase in chlorophyll content compared to control were also observed in Syuhada et al. (2014) in maize leaves sprayed with Cu. Noreen et al. (2021) indicate that the foliar application of Zn in barley cultivation increases the content of photosynthetic pigments, which was also confirmed by the results of our own studies.

The LAI index describes the size of the leaf area per unit of horizontal ground surface. This index is an important biophysical variable to understand the efficiency of the use of radiation by field crops and their potential yield. The LAI value depends on many factors, including genetic traits of plants, their habitat, and agrotechnical factors such as sowing density and the phenological stage of the plant (Chen & Black, 1992; Zou et al., 2018). LAI and MTA indices are primarily used to assess the growth rate and biomass accumulation (Feledyn-Szewczyk, 2009; Lepiarczyk et al., 2005; Dereń et al., 2018; Fang et al., 2019). In our study, the foliar application of each of the microelement fertilizers tested had a positive effect on the LAI and MTA values of barley plants, and the best results were obtained on plots fertilized with Mo (Table 2). Significant genotypic differences were also demonstrated. In the study by Tobiasz-Salach, Jańczak-Pieniążek & Bobrecka-Jamro (2018) in which the foliar application of microelements was used in four barley cultivars, the LAI index also depended on the cultivar and the application of foliar microelement fertilization. There are no reports in the scientific literature on the effect of single-component microelement fertilizers on the MTA value.

Carbon assimilation in photosynthesis and water loss through transpiration are key physiological processes that affect biomass production (Lambers, Oliveira & Pons, 2019; Hatfield & Dold, 2019). The analysis of the effect of fertilization with the microelements tested shows that each of the fertilizers used improved gas exchange parameters in H. vulgare L. The highest increases in values were observed after the application of Cu and Zn. In the conducted experiment, Ci and E were significantly higher after the application of Zn. Zinc plays an important role in the activation of enzymes, N metabolism, and is involved in photosynthesis and DNA reproduction during cell division (Ram et al., 2017). In the studies by Liu et al. (2016), the use of Zn at an optimal dose (30 kg ZnSO4 7H2O) in maize caused an increase in PN, had a positive effect on the chlorophyll content in the leaves, and increased grain yield. Studies conducted on various species of crop plants report that zinc deficiencies in plants cause disorders in the functioning of the photosynthetic apparatus (Mattiello et al., 2015; Rudani, Vishal & Kalavati, 2018; Li et al., 2020). The particularly visible improvement in gas exchange parameters in the experiment carried out after the use of zinc may indicate that the plants were provided with an adequate supply of this nutrient and, consequently, increased photosynthetic efficiency.

In the research conducted, among the microelements tested, the highest Ci and PN values were recorded after the application of Cu (Table 2). The increase in photosynthetic parameters may indicate the optimal dose of Cu for barley plants. In the study by Moraes et al. (2025) on coconut seedlings, in which Cu was applied foliarly at low concentrations, an improvement was observed in the rate of photosynthesis, transpiration, and stomatal conductance. In addition, Cu at lower concentrations increased the efficiency of photosystem II. Copper is potentially harmful when present at concentrations that exceed optimal ones. Excess accumulation can destabilize membrane integrity, reduce photosynthesis, and change enzyme activity, consequently inhibiting plant development (Moustakas et al., 1997; Shabbir et al., 2020; Mir, Pichtel & Hayat, 2021). The study also noted a beneficial effect of foliar Mo spraying on plant gas exchange. Oliveira et al. (2022) showed that the application of Mo to soybeans and corn increased net photosynthesis. The study authors indicated that Mo-based foliar fertilization could effectively improve nitrogen metabolism and plant response to carbon fixation, resulting in improved yields.

The analysis of the values of the chlorophyll fluorescence parameters indicates a particularly significant effect of molybdenum and zinc on the shape of the values of individual indicators. After the application of fertilizers based on these microelements, the highest values of Fv/Fm, RC/ABS, Fv/F0 and PI were recorded (Table 3). Decreases in chlorophyll fluorescence parameters are recorded in plants both under conditions of Zn deficiency (Wang & Jin, 2005) and at too high doses (Andrejić et al., 2018). Studies have indicated a positive effect of foliar application of Zn on the accumulation of photosynthetic pigments and chlorophyll fluorescence, especially in stress conditions (Narimani & Sharifi, 2020; Klofac, Antosovsky & Skarpa, 2023). An increase in Fv/F0 by 6.7% compared to the control was recorded in studies with sugar beet after foliar fertilization with Zn, and the total chlorophyll content increased by more than 25% (Zhao et al., 2024). There are no reports in the literature on the effect of the foliar application of Mo on the chlorophyll fluorescence parameters. Han et al. (2020) investigated the effect of soil Mo application under stress from cadmium (Cd) on rapeseed. In this study, the application of Mo alleviated the negative effect of Cd on plants, among others, by improving Fv/F0 and Fv/Fm.

The response of crop plants to microelement foliar fertilization varies by species and genotype. Barley cultivars, similar to studies by other authors, showed differences in the values of measured physiological parameters (Tobiasz-Salach & Augustyńska-Prejsnar, 2020; Moshfeghi et al., 2019). In a laboratory study conducted by Jarecki, Lachowski & Migut (2024) on soybean where various microelements were applied in foliar form, differences in the response of the cultivars were also shown. In our own studies, a significant increase in CCI, E, and PN was noted in the RGT Planet cultivar compared to the Baryłka and KWS Irina cultivars (Tables 1, 2). In the study by Jalakas et al. (2018) on malted barley, the genotype had a significant effect on the PN and gs values. The chlorophyll fluorescence parameters analyzed from the KWS Irina cultivar showed statistically lower values of the Fv/Fm and Fv/F0 indexes compared to the RGT Planet and Baryłka cultivars (Table 3).

Pearson’s correlation analysis showed the influence of the parameters analyzed on grain yield. The strongest correlations were shown between yield and CCI, E, PN, and PI. A better understanding of the physiological conditions of crop yield formation and the identification of traits related to grain yield help improve the efficiency of crop breeding and cultivation. However, in long-term experiments, physiological parameters are relatively rarely measured in relation to yield (Araus et al., 2008; Jalakas et al., 2018). Jalakas et al. (2018) reported that the correlation between leaf gas exchange traits and grain yield depends on the climatic conditions during the growing season. In studies on wheat and barley, stomatal conductance and CO2 assimilation rate were reported to correlate with grain yield (Fischer et al., 1998; Gonzalez, Bermejo & Gimeno, 2010). In the study by Wasaya et al. (2021), strong positive correlations between PN, gs, and E were observed with grain yield in wheat subjected to drought stress and irrigation. Zhu, Long & Ort (2010) reported that increased photosynthesis in the crop under standard field cultivation conditions resulted in higher yields. Parry et al. (2011) reported an improvement in wheat yield due to increased photosynthesis. Our results also confirm the relationship between the course of plant photosynthesis and the efficiency of barley cultivation.

Conclusions

The studies carried out demonstrated a positive effect of foliar application of fertilizers containing Cu, Mn, Mo, and Zn on the relative content of chlorophyll in leaves, the parameters of the canopy architecture, and the selected parameters of chlorophyll fluorescence and gas exchange in barley plants. The cultivars studied were characterized by a varied response to foliar application of microelements. The values of selected parameters were strongly correlated with grain yield. The analysis carried out indicates that the measurement of physiological parameters during plant vegetation may be useful in forecasting crop productivity. The conducted experiment provides information on the importance of foliar application of micronutrients in the cultivation of an economically important crop species. The research results may be useful in optimizing barley grain production for brewing purposes. Increasing crop efficiency through fertilization without interfering with the soil environment supports the achievement of sustainable development goals. The varied response of the cultivars indicates the need for more research on the optimization of fertilization strategies based on genotypic traits. The research results have significant application implications for precision agriculture and environmentally friendly agriculture. The use of foliar microelement fertilization may contribute to increasing the efficiency of nutrient management, improving the condition of plants under stress conditions, and optimizing yield. This can find practical application in modern crop management systems, supporting farmers in adapting fertilization strategies to changing site conditions and cultivar specificity.

Supplemental Information

Supplemental Information 1 Raw data.

Additional Information and Declarations

Competing Interests

The authors declare that they have no competing interests.

Author Contributions

Barbara Stadnik conceived and designed the experiments, performed the experiments, analyzed the data, prepared figures and/or tables, authored or reviewed drafts of the article, and approved the final draft.

Renata Tobiasz-Salach conceived and designed the experiments, analyzed the data, authored or reviewed drafts of the article, and approved the final draft.

Dagmara Migut analyzed the data, authored or reviewed drafts of the article, and approved the final draft.

Data Availability

The following information was supplied regarding data availability:

The raw measurements are available in the Supplemental File.

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
