# Peer review of "Effect of foliar application of microelements on chlorophyll content, canopy architecture indicators, and physiological parameters of Hordeum vulgare L. plants"

_PeerJ, doi:10.7717/peerj.19966_

## Round 0.1 · original submission · Major Revisions

Dear Dr. Stadnik, I ask you to respond very carefully to each of the reviewers' comments. It is necessary to radically revise this manuscript so that it meets modern world scientific standards.

Reviewer 1 ·

Basic reporting

-

Experimental design

-

Validity of the findings

Impact and novelty not assessed. Meaningful replication encouraged where rationale & benefit to literature is clearly stated.

All underlying data have been provided; they are robust, statistically sound, & controlled.

Conclusions are well stated, linked to the original research question & limited to supporting results.

Additional comments

There is no method part of the study. The study does not contain information on planting norms, maintenance operations, trial installation method, etc.

There are differences between the references in the text and those in the references section. They should be rechecked.

Annotated reviews are not available for download in order to protect the identity of reviewers who chose to remain anonymous.

·

Basic reporting

Comment 1:The manuscript is composed in clear, professional English; however, it contains punctuation and grammatical problems in certain areas, particularly regarding the separation of references by commas and semicolons, which require correction.

Comment 2: The references in the introduction are current and pertinent. The link is expertly constructed.

Comment 3: Figures and illustrations are not available.

Comment 4: Raw data are presented in the annex.

Experimental design

Comment 5: This study is more within the scope of agricultural and environmental sciences and is considered within the scope of the journal, but not within the primary scope.

Comment 6: The research question was well defined, and the answers were given in and at the end of the study. The question is the assumption that foliar application of microelements will contribute positively to the chlorophyll and physiological parameters of barley varieties. This was explained in the study.

Comment 7: Although not very rigorous, the study can answer some questions.

Validity of the findings

Comment 8: The aim of the study and its contribution to the literature are stated.

Comment 9: The data obtained from the study are available and statistically controllable.

Additional comments

Comment 10: Climate and environmental factors should be controlled for each year, and a variety of such studies to get a precise and unambiguous response to the inquiry. However, it is evident that each year's climate data, such as temperature and precipitation, varies. As a result, it is impossible to say whether the change in the measured parameters is due to the microelement applications or the meteorological data. Within the same year, the data can provide clearer answers, but as years pass, the answer becomes ambiguous. Can the authors discuss this situation in detail?

Comment 11: The methods can be given in a more descriptive manner, and the international references on which the methods are based should be indicated as a reference. Method references for both soil analyses and plant parameters should be given.

Comment 12: The results were tried to be answered.

General comments
Comment 13: This study has been previously described as In the article titled ‘’ Influence of Foliar Application of Microelements on Yield and Yield Components of Spring Malting Barley’’ and published at https://doi.org/10.3390/agriculture14030505, yield and yield components obtained from the location and material are presented, while physiological parameters such as chlorophyll content, canopy architecture indicators are presented in this manuscript. Moreover, the climatic data of the location, doses applied, and varieties are the same. Therefore, the following questions need to be answered.

Comment 14: If there is a scientific study on the microelement doses applied, it should be stated and the citation should be written, and other questions; (For example, MIKROVIT® MIEDZ´ 75 (g. L-1) Copper sulfate CuSO4 2 (L. Ha-1) Is there any evidence that this is the optimum dose? Must be explained

Comment 15: Determination of soil physicochemical properties is very important in such studies. However, in the information you gave about the soil in table 3 in the previous study, there is no data on lime ratio, EC and soil texture, and again in table 3, it is seen that the values of copper, manganese, zinc and iron are very variable and abnormal based on years. Can you explain how these values increase disproportionately if sowing was done in the same soil? And you should give a reference to the methods of analysis.

Content of available forms mg·kg-1 soil (Table 3, previous study)

Years
(2019) (2020) (2021)
Copper (Cu) *2.2* 2.4 *3.7*
Manganese (Mn) 164 *396* 225
Zinc (Zn) *5.0* 10.3 *20.0*
Iron (Fe) *1080* 1563 *3824*

Comment 16: In addition, in your previous study (Table 3) in the soil analysis table, there is no data on molybdenum, but there is data on iron; you could have used iron as a dose. Can you explain why you eliminated iron? However, Fe concentration in the soil analysis was very variable and high. Have you considered that this may affect the results every year?

Comment 17: In addition, it is mentioned in the sources that the best intake range of micronutrient elements in the soil is between 6,5-7,5. However, in your table 3, the pH range is between 5,8-6,0, that is, acidic soil, although you have used sulphur, this increases the soil acidity even more. Soil texture is also an important parameter. It gives an idea, especially for the water holding capacity of the soil, clay soils hold a lot of water while sandy soils hold the least water, which is important for the difference in rainy and dry years. Temperature is the most important factor affecting evaporation.

Comment 18: In this study, the significance factor P is not specified in the correlation table 4, and which correlation coefficient is used is not stated, it would be better if the coefficients (Pearson or Spearman) are expressed as a star on the upper right side of the coefficients (i.g. 0.62** ), and if the correlation of all parameters is tabulated, perhaps it would be more useful for the scope of the study.

·

Basic reporting

Thank you to the Editor for the opportunity to review this manuscript. The study addresses the important topic of improving the physiological efficiency of barley cultivation through the application of foliar micronutrients. The authors have conducted a multi-year field experiment which, in principle, merits recognition.

Introductory comments (lines 49–53):
The manuscript rightly highlights foliar feeding as an environmentally friendly and well-controlled method of delivering nutrients. However, it should also be noted that foliar fertilisation is particularly valuable when root uptake is impaired, for instance due to drought, soil compaction, or oxygen deficiency. Including this remark would reinforce the physiological rationale, particularly with regard to the high-yielding crops mentioned in lines 52–53.

Micronutrient requirements (lines 58–59):
The statement 'Barley requires micronutrients such as copper (Cu), manganese (Mn), molybdenum (Mo) and zinc (Zn) for optimal growth and development' (Nyiraguhirwa et al., 2022) does not reflect the variability of micronutrient limitations across edaphic–climatic conditions. The cited source may stem from a regional study; in other agro-ecosystems, different elements (e.g. boron or iron) may be critical. Please nuance this statement accordingly.
Descriptive section (lines 64–91):
This section largely repeats information found in textbooks and does not substantiate the scientific gap that the study intends to fill. I recommend either trimming or rewriting the section to focus on unresolved questions about how foliar micronutrients affect barley physiology, as this is where true novelty lies.

Experimental design

Redundancy (line 97):
The phrase 'a three-year (2019–2021) field experiment' contains a redundant repetition of the years. One mention is sufficient.
Field experiment description (lines 97–113):
The current description is too brief to ensure reproducibility. Please specify:
• plot size;
• the number of replicates;
- randomisation/blocking scheme;
- soil properties, climatic conditions, sowing date and density, basal N-P-K fertilisation, etc.
The list of foliar products (lines 107–111) is cumbersome. Consider using a table with the following columns: product name, active ingredient, chemical form, dose, and application method. The 'Field experiment' subsection needs substantial expansion to satisfy PeerJ transparency standards.
BBCH phases and averaging (lines 117–118):
BBCH 43–45 and 50–52 are cited without explanation. Please provide parenthetical definitions. Furthermore, averaging physiological data across these distinct phenological stages (late stem elongation vs. early heading) requires justification or separate analyses to show that the dynamics between stages did not bias the conclusions.

Statistical analysis (lines 151–157):
Although ANOVA (split-block) and Tukey HSD are mentioned, essential details are missing.
- which fixed and random factors were fitted;
- whether year effects were included;
- assumption checks (normality and homoscedasticity).
- whether analyses were performed per year/cultivar/micronutrient or pooled.
The correlation analysis lacks information on yield determination, the structure of observations, the replication level and whether variability across years/genotypes was considered. Please clarify:
1. The factor structure in ANOVA;
2. How year and cultivar variability were treated;
3. Yield measurement methodology and aggregation level;
4. Diagnostic tests of ANOVA assumptions.
Tables 1–3 (descriptive vs. effect estimates):
The tables present means but omit:
• sample sizes (n);
• ANOVA F-statistics, degrees of freedom and p-values.
- clarity on which factor levels underpin Tukey groupings;
The tables are also densely formatted and lack a readable layout.
Therefore, while the tables illustrate the data, they do not rigorously support the stated effects.
Table 4 (correlations):
Major issues:
1. There is no sample size (n) per correlation coefficient.
2. There are no confidence intervals or p-values.
3. There is no test of homogeneity across years, cultivars or treatments (Fisher z-tests).
4. The aggregation level is unclear (year means, genotype × treatment means or individual plots?).
5. The abbreviations (CCI, LAI, MTA, Ci, E, gs, PN, Fv/Fm, RC/ABS, Fv/F0 and PI) are not explained.

Validity of the findings

The validity of the findings is questionable due to numerous methodological uncertainties. The manuscript is deficient in the provision of detail concerning the experimental design, including aspects such as randomisation, replication, and plot size. Additionally, there is a paucity of information regarding the treatment structure and the sample size. The statistical analysis is inadequately described, with effect estimates and variance measures absent. Correlations are presented without significance testing, variability estimates or robustness checks across years, cultivars or fertilizer treatments. These shortcomings prevent a rigorous assessment of the reliability and generalisability of the results; therefore, the validity of the findings cannot be confirmed in the present form.

---

## Round 0.2 · accepted · Accept

Dear Dr. Stadnik, I congratulate you on the acceptance of this article for publication and hope that you will continue your research, which is important for the food security of all mankind. I wish you success in your scientific work and many, many more such good publications.

The Section Editor noted that:

> Table legends for tables 2, 3, and 4 are unwieldy with the long list of df, F, and p-values. These should be incorporated into the table (or into a separate table if necessary).

·

Basic reporting

The article is written in clear English.

Experimental design

The experimental part has fulfilled the requirements of the study.

Validity of the findings

The results are expressed as accurately as possible.

Additional comments

Hello,
The article has been revised taking into account most of the points we mentioned earlier. Therefore, the article is acceptable.

·

Basic reporting

The authors have implemented all the recommendations of the reviewer. The quality of the manuscript has been significantly improved. I recommend the article for publication.

Experimental design

The authors have implemented all the recommendations of the reviewer. The quality of the manuscript has been significantly improved. I recommend the article for publication.

Validity of the findings

The authors have implemented all the recommendations of the reviewer. The quality of the manuscript has been significantly improved. I recommend the article for publication.